# Exploring the spatial working memory and visual perception in children with autism spectrum disorder and general population with high autism-like traits

**Manxue Zhang[1], Jian Jiao[1], Xiao Hu[1], Pingyuan Yang[1], Yan Huang[1], Mingjing Situ[1], Kuifang Guo[1], Jia Cai[1], Yi Huang** [1,2,3] *

1 Mental Health Center, West China Hospital of Sichuan University, Chengdu, China, 2 Brain Research Center, West China Hospital of Sichuan University, Chengdu, China, 3 Psychiatric Laboratory, State Key Laboratory of Biotherapy, West China Hospital, Sichuan University, Chengdu, Sichuan, China

* huangyi0412@126.com

**Data Availability Statement:** All relevant data are within the manuscript and its Supporting Information files.

## Abstract

The aim of the study is to compare the spatial working memory and visual perception between children with autism spectrum disorder (ASD) and typically developing control (TDC). Furthermore, this study validated whether this impairment was a feature of autism in general population with different autism-like traits (ALTs). This study contains two parts: case-control study and community population study. The ASD group and the control group were enlisted voluntarily (ASD group, n = 52; control group, n = 32). In the population study, we recruited 2994 children. Based on the scores of Autism Spectrum Quotient (AQ), children were divided into two groups (higher ALTs n = 122, lower ALTs n = 122). The participants completed the cognition tasks focusing on spatial working memory, visual-motor integration, and Intelligence. Analysis of covariance (ANCOVA) was conducted, with potential confounders IQ, age, and gender were controlled. Pearson correlations were computed by controlling the IQ and age as covariate to better understand the relations between visual perception, spatial working memory, and autism-like traits. In the case-control study, the results of cognition tasks focusing on the spatial working memory and visual perception indicated underperformance in children with ASD. In the community population study, we found that individuals with higher ALTs performed worse than children with lower ALTs in spatial working memory. Pearson correlation analysis suggested that a correlation between SWM total errors and visual perception was identified both in the children with ASD and in community population (ASD group, r = -0.592, p<0.001; general population, r = -0.201, p = 0.003). It suggested that spatial working memory deficit was a characteristic of autism, and may be distributed across the general population. Furthermore, we speculated a correlation between spatial working memory and visual perception in children with ASD and in general population.

**Funding:** This work was supported by the National Key Research& Development Program of China (NO.2016YFC1306100) and The National Natural Science Foundation of China (NO.81371495).

**Competing interests:** The authors have declared that no competing interests exist.

## Introduction

Autism spectrum disorders (ASD) is a neurodevelopmental disorder, characterized by deficits in social interaction, social communication, and restricted, repetitive patterns of behavior, interests including unusual sensory perceptions [1]. Perceptual symptoms such as visual reception, auditory and motor are prominent in the early life of ASD [2, 3] and show a persistent relationship to clinical measures of cognition and behavior [4]. Mental processes are the way people perceive the world, and sensation, perception, and cognition probably affect one another to different degrees.

Visual perception was identified as the ability to receive, recognize, analyze, and elaborate the visual stimulus from objects and events [5]. As early as 28 weeks, most infants can learn to distinguish between a circle, square, and triangle [6]. Part-whole integration has been of special significance in visual perceptual development, while many children with ASD seem to be unable to synthesize parts into the whole [7, 8]. Visual problems and spatial visual processing dysfunction in ASD might vary in onset, severity and behavior patterns. They can be generally grouped into hypersensitivity and hyposensitivity. Previous studies have shown that visual perceptual dysfunctions in early age would be predictive of future ASD symptoms over time [9–12]. Visual processing is a complex process, including visual recognition, visual memory, visual spatial orientation, and the perception of graphics. Several investigations showed that the results of visual perception are inconsistent, as evidenced by impairment of eye movements and contrast sensitivity [13, 14]; enhanced visual searching [15, 16] and hyper-local orientation [17] in individuals with autism; normal in orientation processing, crowding, and flicker detection [18–20]. The inconsistences may be explained by different tools used to exploring various aspects of visual processing. The present study focuses on overall visual perception. In addition, the inconsistency of visual perception studies results might be attributed to the heterogeneity of ASD, it is necessary to study from the perspective of the continuous distribution characteristics.

Previous researches had provided inconclusive evidence about whether children with ASD are impaired in working memory. Investigations of school aged children and adolescents suggested working memory impairment in ASD [21, 22]. Working memory deficit is associated with symptoms of children and adolescents with ASD such as communication and socialization impairments [23, 24], restrictive and repetitive interests and behaviors [25]. While some researches indicated normal working memory in preschoolers and adults with ASD [26–28]. The mixed results indicated that adolescents with ASD showed remarkable working memory dysfunction, and indicated that age had important influences in the field. A Meta-Analysis [29] of working memory in individuals ranged from school-age children to adults suggested that spatial working memory was more severely impaired than verbal working memory. Several studies investigated the spatial working memory to confirm the dysfunction. However, whether children with ASD are impaired in the spatial working memory was not confirmed because of different measures used [21, 30, 31]. Furthermore, the visual perception requires executive functions such as planning and working memory, visual perception also affects the test results of spatial working memory.

We believe cognitions should be seen as complementary rather than mutually exclusive. There have been many theories about the psychopathological mechanisms of ASD, such as 'executive function deficit', and 'weak central coherence' [32, 33], but none of these theories can explain the whole picture of the symptoms of ASD. Neurobiological accounts of how and why these lower and higher order symptoms might be related in autism are largely divided into two camps: Sensory first accounts [34] and Top down accounts [35–37]. There are many studies on spatial working memory and visual perception, but there are few studies on the

relationship between them, and the results are inconsistent: Mottron and his group [35] put forward a general tendency toward a local bias of autism that may be related to impairments in the maintenance of several representations in spatial working memory. However, the investigation did not explore the interaction between them. One study found abnormalities in executive function and visual motor integration in premature infants, and the two cognitions were moderately correlated [38]. Another study investigated in young healthy children have shown similar correlation between working memory and visual motor integration [39]. But the results in preschool children suggest that working memory does not affect visual perception [40]. Karisa and his team, using a sophisticated measure that tested visual processing and executive function, found that visual perception is dissociable with executive function in individuals with various degree of ALTs [41]. Further experiments will be necessary to discover the direct relationship between visual perception and spatial working memory in the processing of visual information.

Autistic-like traits (ALTs) were defined as subclinical traits in the broader population and were continuously distributed throughout the normal range to the clinical extreme population [42]. Individuals with high ALTs show similar symptoms [43, 44] and cognition deficits such as, visual-spatial process [45–47], working memory [48], shared-attention [49], and cognitive flexibility [50]. Research indicated that the majority of genetic liability for ASD is attributed to common inherited variances [51]. Therefore, the ALTs exist continuously in the general population. There are several methodological advantages that general population samples brought to the research on the etiology of autism, such as substantially more power to conduct model-fitting analyses (see review [52]).

The aim of the study is to investigate the visual perception and spatial working memory function deficits in children with ASD compared with typically developing controls. Meanwhile, we are interested in whether the results would be repeated in population study when comparing higher trait individuals with lower trait individuals. Furthermore, we detected the correlations between visual perception and spatial working memory.

## Materials and methods

### Participants

This study contains two parts: case-control study and community population study. We recruited ASD without intellectual disability comorbidity(n = 52, mean age = 9.23 years, SD = 3.35 years, Min–Max = 4–17 years; mean IQ = 95.27,SD = 19.18, Min–Max = 70–132; Male/Female = 37/7) and typically developing control children (n = 32, mean age = 10.63 years, SD = 3.15, Min–Max = 6-16years; IQ = 108.22, SD = 13.94, Min–Max = 81–136; Male/Female = 16/16). (see in Table 1) during the period from September 2016 to June 2018. Both off-line and on-line recruitment methods were adopted in the recruitment process. Off-line method were conducted in the West China Hospital outpatient. Our online recruitment advertises were on the main social platforms including WeChat as well as the official accounts of the schools and hospitals involved. Participants were included after evaluation based on Autism Diagnostic Interview-Revised (ADI-R), Autism Diagnostic Observation Scale-General (ADOS-G), and clinical judgment by experienced child psychiatrists.

We also investigated in the sites of three primary schools in Pi Xian county Xi Pu District, in the city of Chengdu, Sichuan Province, China. Combination of the following factors was considered to select the district: 1) population from 5,000 to 15,000; 2) the lowest rate of the migrant population in the past years, 3) cooperation from the local government. The eligible children were identified in each selected street through the local public security bureau household registration system. Pi Xian is a county with 750,000 population consists of 13 districts.

**Table 1. Participants demographics characteristics.**

| Measures | | ASD group(n = 52) | TDC group (n = 32) | t/χ² | p |
|---|---|---|---|---|---|
| Gender ratio (Male/Female) | | 37/7 | 16/16 | 10.2 | 0.001** |
| Age (year) | Mean | 9.23 | 10.63 | 1.896 | 0.062 |
| | SD | 3.35 | 3.15 | | |
| | Min | 4 | 6 | | |
| | Max | 17 | 16 | | |
| IQ | Mean | 95.27 | 108.22 | 3.191 | 0.002** |
| | SD | 19.18 | 13.94 | | |
| | Min | 70 | 81 | | |
| | Max | 132 | 136 | | |

*p<0.05

**p<0.01, ***p<0.001; ASD: autism spectrum disorders; TDC: typically developing controls; IQ: Intelligence Quotient

The population study was conducted in Pi Xian county, Xi Pu district from October 2016 to January 2017. Two thousand nine hundred ninety-four school aged students in three public primary schools -Si Yuan experimental primary school, Xi Pu experimental primary school, and Xi Pu foreign language primary school- were included in our study which can stand for the characteristics of the children population of the district. According to the sample size calculation formula N = $(U\alpha S/\delta)^2$, ($U\alpha$ is the U value corresponding to the inspection level $\alpha$, set as $U\alpha = 1.96$; S is the standard deviation, $\delta$ is the allowable error; assume $S/\delta = 5$), the estimated sample size is N = 99. Considering the loss of interview and quality control, we chose all the students in the schools. All the 2994 students studying in the schools were recruited in this investigation. The caregivers of these children were asked to fill in the questionnaire: The Autism Spectrum Quotient (AQ). After quality control of the questionnaires filled by parents, the ultimate response rate was 70.33%. According to the AQ score, 15% of the high score was set as the higher trait group, including a total of 122 individuals. Then, using stratified random sampling method take samples of 122 children from the remaining people as the lower trait group.

All the participants were asked to complete cognition tasks including the SWM tasks, the Berry VMI tasks, and the short-form Chinese Version of the Wechsler Intelligence Scale. The exclusion criteria of our study included: a history of psychotic disorder, severe head injury, syndromic genetic disorder associated with autism (e.g. fragile X syndrome), intellectual disability (i.e. IQ <70), Tourette's disorder, attention deficit hyperactivity disorder, any other medical condition significantly affecting brain function; unable to cooperate to complete the test. Finally, the demographic features of the population study participants were as follows, higher trait group(n = 115, mean age = 8.35 years, SD = 1.42 years, Min–Max = 6–11 years; mean IQ = 117.37, SD = 12.25, Min–Max = 86–139; Male/Female = 44/71), lower trait group (n = 105, mean age = 8.03 years, SD = 1.50 years, Min–Max = 6–11 years; mean IQ = 116.16, SD = 10.74, Min–Max = 82–137; Male/Female = 42/63).(see in Table 3) Parents of all the participants provided the written informed consent for study participation, which was approved by the Medical Ethical Committee of West China Hospital of Sichuan University.

## Measures

**Autism Spectrum Quotient (AQ).**   The original AQ is a 50-item self-administered questionnaire for the explicit purpose of measuring tendency towards autistic traits [53]. Cultural study had been investigated in several countries and showed good property. Winnie Yu-Pow

Lau et al [54] adapted abridged AQ version in Chinese samples, consisting of 35-item. AQ-35 Chinese version consisted of five tightly semantically coherent subscale constructs, named as Socialness, Mindreading, Patterns, Attention to Details and Attention Switching. The items clustered in a way that depicted distinctive dimensions of ASD symptomatology.

**Autism Diagnostic Interview-Revised (ADI-R) and Autism Diagnostic Observation Scale-General (ADOS-G).** ADI-R is used for children aged from approximately 18 months to adulthood [55]. The Chinese ADI-R was approved by the World Psychological Association in 2007 [56]. The ratings were based on an assessment under current conditions and under the most severe state at 4–5 years, as recalled by the caregivers. It covers most developmental and behavioral aspects of ASD, including reciprocal social interaction, communication, and repetitive behaviors and stereotyped patterns. The ADOS-G is a semi-structured, standardized assessment of social interaction, communication, play, and imaginative use of materials [57]. It has four different units for different levels of development and verbal abilities. ADOS-G provides scores that are distinct in three domains: language and communication score, reciprocal social interaction scores and total scores. ADI-R and ADOS-G are the golden standards for ASD diagnosis.

**Chinese version of the Wechsler Intelligence Scale (short-version).** Wechsler Intelligence Scales for Children-III (WISC-III) [58] was used to measure the intelligence of children. In this study, we used the Chinese short-version of the Wechsler Children Intelligence Scale-III, which consists of Vocabulary, Similarity, Picture Completion, and Block Design. For the present investigation, FSIQ was measured using the WISC–III.

**Spatial Working Memory (SWM).** The Cambridge Neuropsychological Test Automated Battery CANTAB eclipse Test (Administration Guide Manual version 5.0.0 http://www.cambridgecognition.com/) is a set of computerized paradigms run on a Lenovo-compatible computer with a high-resolution color monitor and the touch-sensitive screen. In our study, children were requested to complete Spatial Working Memory (SWM). Kids were asked to search through a number of colored boxes presented on the screen to find blue tokens hidden inside, which is set to be shown only once a trail in each box. Touching any box in which a blue token has already been found is an error. The number of boxes is gradually increased from three to eight boxes. The color and position of the boxes used are changed from trial to trial to discourage the use of stereotyped search strategies. The outcome measures for the SWM test may be regarded as "Total Errors" in previous studies. The lower the total errors of SWM, the better the cognitive function of Spatial Working Memory. The SWM, based on a self-ordered search test [59], assesses spatial working memory. Also, the SWM had been used to assess the spatial working memory among ASD children [21, 30, 31].

**The Developmental Test of Visual-Motor Integration (Beery VMI).** The Developmental Test of Visual-Motor Integration (Beery VMI) [5, 60] is a developmental sequence of geometric forms to be reproduced with paper and pencil and scored according to objective scoring criteria outlined in the test manuals based on accuracy of copying when compared to the original. Considering the longitudinal nature of the parent project, the 3-5th editions of the Beery VMI were used during the various stages of data collection. The stimuli in the various editions did not change. In the subtest of visual perception, children need to choose the figure which was same as the one in the box above, one point is awarded for each correct item until three consecutive incorrect item scores or the 3-minute time limit expires. The Beery VMI has been demonstrated to have good reliability and validity as reported in the manual. Beery VMI is one of the most commonly used standardized measurements of visual-motor integration, visual perception and motion coordination in several developmental disorders. The Beery VMI has a rich tradition in assessing children and adults with various neurodevelopmental disorders, including ASD [61–63].

### Statistical analysis

Statistical analyses were carried out with the Statistical Package for the Social Sciences, 22.0 (SPSS 22.0). The statistical significance level was set to p ≤0.05 (two-tailed) for all tests. Independent samples t-tests and Analysis of covariance (ANCOVA) were conducted to compare groups of demographic information and cognitive factors, with potential confounders age, gender, and IQ was controlled. Pearson correlations were computed under age, IQ control to better understand the relations between Beery VMI, SWM performance and AQ scores.

## Results

### Case-control study

**Participants demographics characteristics.** We chose children with ASD whose IQ scores were higher than 70, to avoid the intelligence influences on cognition tasks. The average IQ of ASD group was normal, IQ = 95.76, although there was a significant IQ difference between groups (t = 3.191, df = 83, p = 0.002). The t-test of independent sample results indicated that there is no significant difference in age between the ASD group and the TD control group (t = 1.896, df = 83, p = 0.062). And the rate of genders was especially unbalanced in the ASD group ($\chi^2$ = 10.2, df = 83, p = 0.001). (see in Table 1)

**Group differences regarding neuropsychological test performance.** The t-test results showed significant differences between ASD group and TDC controls in all cognitive functions tested in our study except for visual perception. However, to minimize the effect of unmatched IQ, age, and gender between groups, the ANCOVA was conducted with IQ, gender, and age as covariant. Results of cognition tasks focused on the spatial working memory showed significant group differences (F = 2.986, df = 83, p = 0.004), which indicated a spatial working memory dysfunction in ASD individuals. We also found children with ASD performed worse in visual perception (F = 2.006, df = 83, p = 0.048), motion coordination (F = 3.416, df = 83, p = 0.001) and visual-motor integration (t = 2.150, df = 83, p = 0.035), which may indicate visual perception and motor dysfunction in ASD individuals. (see Table 2)

**Correlations among neuropsychological tests and symptom severity.** We performed Pearson correlation analysis among neuropsychological test performances in ASD group using IQ, gender, and age as covariant. The correlations between SWM, VMI and AQ score were not significant (p>0.05). We further explored the correlations between SWM and VMI tests. There was negative correlation between SWM total errors with visual perception (r = -0.592,

**Table 2. Performance differences on Spatial Working Memory (SWM) and Berry-VMI between ASD and control.**

|  | ASD(n = 52) | TDC (n = 32) | t | p | F^c | p^c |
|---|---|---|---|---|---|---|
| AQ | 19.65±3.71 | 13.97±4.80 | 6.083 | <0.001*** | 6.083 | <0.001*** |
| SWM total errors | 57.81±16.78 | 45.88±19.32 | 2.086 | 0.002** | 2.986 | 0.004** |
| VMI-Integration | 96.65±19.28 | 104.88±12.43 | 2.150 | 0.039* | 2.150 | 0.035* |
| VMI-Visual | 102.00±19.04 | 109.44±11.13 | 1.549 | 0.123 | 2.006 | 0.048* |
| VMI-Motion | 95.46±15.79 | 106.97±13.58 | 3.504 | <0.001*** | 3.416 | 0.001** |

*p<0.05

**p<0.01

***p<0.001. F^c: F value after corrected with IQ,gender, and age as covariant; p^c: p value after corrected with IQ, gender, and age as covariant; ASD: autism spectrum disorders; TDC: typically developed controls; AQ: Autism Spectrum Quotient; SWM: Spatial Working Memory; VMI-Visual: The Developmental Test of Visual-Motor Integration-visual perception subtest.; VMI-Motion: The Developmental Test of Visual-Motor Integration-motion coordination subtest; VMI-integration: The Developmental Test of Visual-Motor Integration-integration subtest.

**Table 3. Participants demographics characteristics.**

| Measures | | H-trait group (n = 115) | L-trait group (n = 105) | T/$\chi^2$ | p |
|---|---|---|---|---|---|
| Gender ratio (Male/Female) | | 44/71 | 42/63 | 0.069 | 0.79 |
| Age (year) | Mean | 8.35 | 8.03 | 1.619 | 0.107 |
| | SD | 1.42 | 1.42 | | |
| | Min | 6 | 6 | | |
| | Max | 11 | 11 | | |
| IQ | Mean | 117.37 | 116.16 | 0.562 | 0.575 |
| | SD | 12.25 | 10.74 | | |
| | Min | 86 | 82 | | |
| | Max | 139 | 137 | | |

*p<0.05, **p<0.01, ***p<0.001; H-trait group was abbreviation for higher trait group; L-trait group was abbreviation for lower trait group; IQ: Intelligence Quotient.

p<0.001), and motion coordination (r = -0.398, p = 0.004). (Fig 1) It indicated that spatial working memory dysfunction was related with visual perception in children with ASD.

In the mean time we conducted Pearson correlation analysis in healthy control children. Results showed that there was no correlation among spatial working memory, visual perception and ASD symptom severity(p>0.05). (Fig 2) The results showed the correlation between SWM and visual perception in children with ASD did not repeat in typically developing controls.

## Community population study

Community population study was conducted in three sites, after stratified random sampling and quality control the final participants demographics characteristics were as follows, higher trait group n = 115,lower trait group n = 105.There was no significant differences between the two groups in age, gender, and IQ (p>0.05).(described in Table 3.)

**Neuropsychological test performance differences among different degrees of autistic-like traits.**    Independent t test results revealed that individuals with higher ALTs behaved worse than children with lower ALTs in spatial working memory. (t = 2.117, df = 119, p = 0.035) We did not find group differences in visual perception. (p>0.05) (see Table 4 for descriptive statistics).

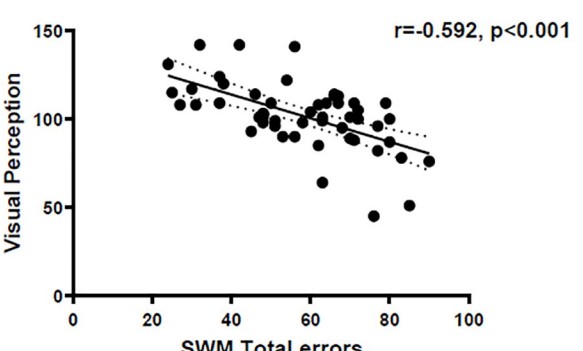 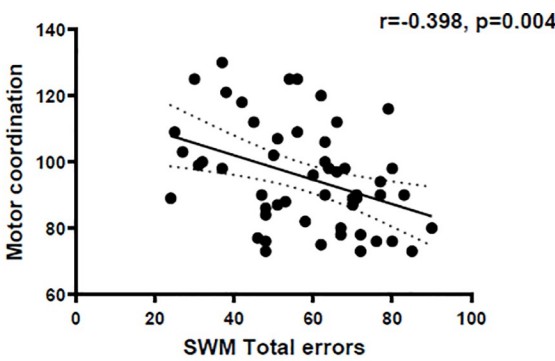

**Fig 1. Correlations between Spatial Working Memory and VMI performances in children with ASD.** The Pearson correlation analyses were conducted with age and IQ as covariant. The left picture showed a correlation between spatial working memory and visual perception (r = -0.592, p<0.001); The right figure indicated a correlation between spatial working memory and motion coordination (r = -0.398, p = 0.004).

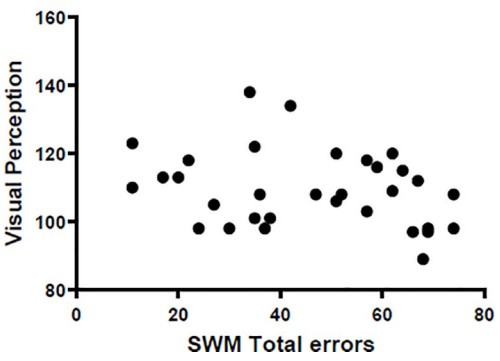 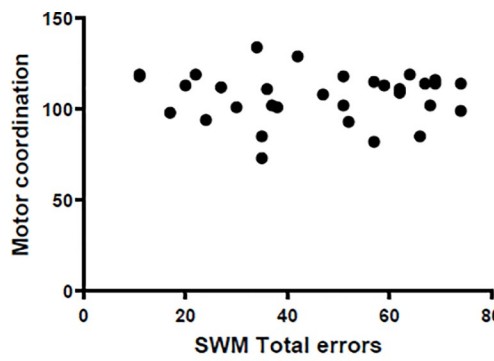

**Fig 2. Scatter diagrams of Spatial Working Memory and VMI performances in typically developing children.** The Pearson correlation analyses were conducted with age and IQ as covariant. The left picture showed the correlation between spatial working memory and visual perception was not significant ($r = -0.214$, $p = 0.256$). The right picture showed the correlation between spatial working memory and motor coordination was not significant ($r = 0.099$, $p = 0.604$).

**Correlations among neuropsychological tests and autistic traits.** We performed Pearson correlation analysis to determine the association between neuropsychological test performances and ASD Symptoms in the community population samples. There was negative correlation between SWM total errors with VMI scores (SWM total errors and VMI-visual perception: $r = -0.210$, $p = 0.003$; SWM total errors and VMI-motion coordination: $r = -0.138$, $p = 0.041$). (see in Fig 3) The correlations between SWM, VMI and AQ score were not significant ($p > 0.05$).

## Discussion

This study found that children with ASD had abnormalities in spatial working memory, visual perception compared to typically developing controls. Furthermore, the results indicated a correlation between the cognitions in the ASD group. Impairment of spatial working memory in children with ASD has been demonstrated in population study, with children with high ALTs performing worse on spatial working memory than children with low ALTs. The correlation between visual perception and spatial working memory was confirmed in general population.

Working memory was defined as a limited capacity system that maintains information "on-line" over brief periods of time [64]. Spatial working memory is responsible for the temporary storage of visual information, and its "on-line" manipulation. This study found the spatial working memory deficits in both children with ASD and children with high ALTs, which is consistent with previous studies. Studies indicated that children with ASD had dysfunction

**Table 4. Tasks performance differences between different trait group on Spatial Working Memory (SWM) and Berry-VMI.**

|  | H-trait group(n = 115) | L-trait group(n = 105) | t | p |
|---|---|---|---|---|
| SWM total errors | 49.70±19.35 | 44.02±20.48 | 2.117 | 0.035* |
| VMI-Integration | 102.16±13.69 | 103.23±13.17 | 0.591 | 0.555 |
| VMI-Visual | 107.47±11.87 | 109.61±12.54 | 1.300 | 0.195 |
| VMI-Motion | 100.60±16.35 | 101.96±13.61 | 0.668 | 0.505 |

*$p < 0.05$, **$p < 0.05$, ***$p < 0.001$; H-trait group was abbreviation for higher trait group; L-trait group was abbreviation for lower trait group. IQ: Intelligence Quotient; SWM: Spatial Working Memory; VMI-Visual: The Developmental Test of Visual-Motor Integration-visual perception subtest.; VMI-Motion: The Developmental Test of Visual-Motor Integration-motion coordination subtest; VMI-integration: The Developmental Test of Visual-Motor Integration-integration subtest.

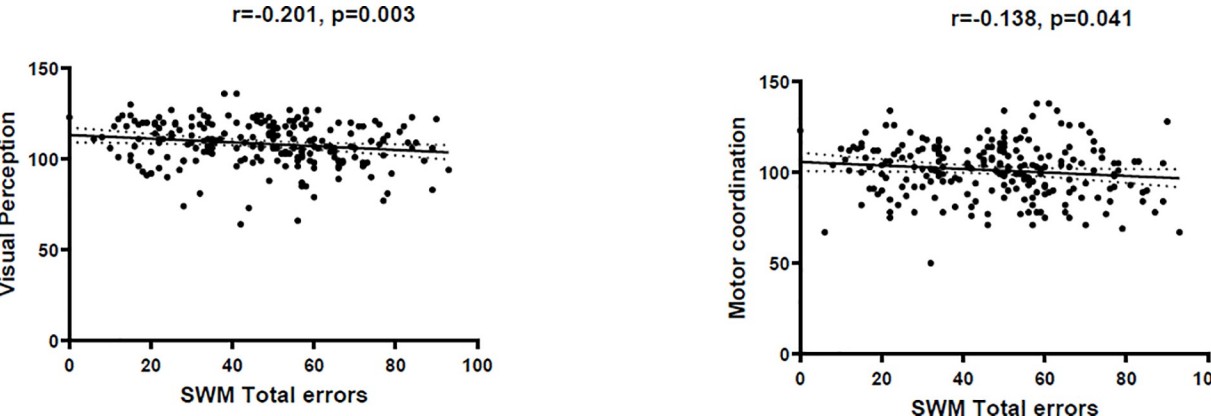

**Fig 3. Correlations between spatial working memory and VMI performances in community population.** The Pearson correlation analyses were conducted with age and IQ as covariant. The left picture showed the correlation between spatial working memory and visual perception (r = -0.201, p = 0.003). The right picture showed the correlation between spatial working memory and motor coordination (r = -0.138, p = 0.041).

in the spatial working memory [21, 22]. Spatial working memory abnormalities were found to be correlated with autism symptom severities in previous study [23]. On contrary, we found that the association between SWM and AQ was not statistically significant. It may be related to different age ranges. Previous studies were conducted in a wide range of age from children to adolescents, and our study was conducted in children aged from 6 to 12 years. Secondly, the tools used are different. A questionnaire, the Behavior Rating Inventory of Executive Function, BRIEF was used in the previous study; however, the present study used CANTAB which was a common and reliable measure. The BRIEF is not specific enough to capture separate EFs based on the subscales [65]. Furthermore, symptoms may be affected by many factors, and the effect of cognitive function on symptoms may be related to other mediating factors. Interaction between cognitive functions may also have an impact on the results that SWM impairment found in ASD and high ALTs is not associated with overall level of autism symptom severity measured with the AQ. Another possible explanation for this pattern could be that spatial working memory is related to aspects of autism that may not be covered well by the AQ. Additional studies using larger samples of participants in all age periods that combine laboratory and informant-based measures are necessary to clarify this issue. However, recently one study conducted in unaffected siblings, showed spatial working memory wasn't a broader autism phenotype of ASD [48]. The previous study didn't identify spatial working memory as broader autism phenotype of ASD, which may be explained by the small sample size and the limitation of the term 'broader autism phenotype'; and in our study we applied AQ to measure ALTs. Children with high ALTs have also been found to have spatial working memory deficits [66]. Furthermore, Previous neurobiological studies using electroencephalograph (EEG) [67] or fMRI [21] results suggested that children with ASD have abnormal activation of brain regions associated with working memory. Investigating executive function components that contribute to the severity of autistic symptomology provides insight for the development of effective interventions for individuals with ASD. Previous study proposed that the working memory trains improved classroom performances of children with ASD [68]. We speculated that spatial working memory was an autistic-like trait; and professional spatial working memory training might alleviate the severity of the symptoms.

Abnormal sensory perception is one character of autism. However, previous studies on visual processing had inconsistent results due to different focus points [13, 15–17, 69]. This study focused on overall visual perception. In our study, the visual perception function deficit

was found in ASD group when compared with typically developing children. Previous studies have found that individuals with autism have visual perception dysfunctions, which was predictor to the severity of symptoms [70–75]. Except that Green et al. did not repeat the results in children with ASD [63]. However, the population study results indicated that higher trait group behave comparable of lower trait group. This is at odds with previous research suggesting that people with high autistic traits have impaired visual illusions [76], and visual search [77]. It should be noted that the AQ also correlates with IQ [78], and it could be IQ that produces the no significant results of higher trait group. In addition, the visual perception inconsistency of the ALTs prompted us to carry out better visual psychophysics using more sensitive measures of visual perception in general population with various degrees of autistic traits. Future brain mapping studies could provide additional insight into the neural underpinnings of how visual perception might and might not be affected in autism.

It is challenging for traditional case-control studies to invest diseases that distributed normally in population. However, population study is just helpful to explore the diseases with a continuous distribution of characteristics in the population [79]. Therefore, in the current study, we investigate the spatial working memory and visual perception by using a general population approach. Besides comparing a group with a clinical diagnosis of ASD to a group with typical development, we also compare individuals with higher ALTs to individuals with lower ALTs. ASD is commonly operationalized as the extreme end of a phenotypic continuum of autistic traits normally distributed in the general population [80]. Studies found that high ALTs individuals have behavioral and cognitive characteristics similar to those of ASD [81–83]. This study consists of two parts, case control study and population study. The results confirmed that the high ALTs had similar changes in neurocognitive function as in children with ASD, suggesting that spatial working memory may be a core symptom of ASD. Besides, the present study offered support to the approaches that regard autistic-like traits as a continuous variable in general population rather than qualitative variable.

Studies have shown that working memory and other executive functions are important contributing factors influencing visual processing performance [38, 39]. When the subjects were asked to complete the VMI visual perception test, they require the ability to visualize an image, hold the image in short-term memory, and transfer the image with the correct proportions onto paper. The present study found that visual perception deteriorated as working memory performance decreased in the ASD group. While the relationship didn't repeat in control group, possibly because copying geometric figures in the VMI tasks was not so hard for typically developing children to remember; at the same time children could check the geometric figures again if they forgot them. Also, the small sample size may have an impact on the results of control group, and we tried to address it by community population study with a larger sample size, and results of the general population children suggested a positive correlation between SWM and visual perception. Researches in preschool children suggested working memory has no impact on the VMI skills [40], the age range might explain the controversy. Further study was needed to explore mechanism of the relationship between working memory and visual perception using more meticulous measures. Neuroimaging studies found that executive function and visual perception are connected by overlapping neural networks in individuals with ASD [84–86]. Further intensive structural equation model is needed to clarify the relationship between visual perception and spatial working memory, and to find out the underneath visual processing mechanism of neuropsychology in ASD.

Our study design includes population-based study that increase the effectiveness of finding core traits of ASD. The spatial working memory deficits found both in children with ASD and in ALTs individuals reflect that the cognitive characteristic may be distributed across the general population. Furthermore, we investigated the correlation between spatial working

memory and visual perception, attempting to verify the mutually relationship between different cognitions in visual processing. The weakness of the study was that the gender ratio of the clinical samples was unbalanced which will have some impact on the final results when concerning the cognitive function impairments. The situation may be caused by a higher incidence of boys. When compared ASD with TDC in the cognitions, gender was set as a covariant to control potential bias. Another disadvantage is that the instrument we used, the VMI, cannot describe the visual processing in detail. In the future, more elaborate task design is needed to explore the visual processing of autism. Finally, our study is a cross-sectional study, long-term follow-up studies can be conducted in the future, considering the impact of development on children.

## Conclusions

The findings of our study indicated spatial working memory played an important role in the ASD symptoms. Our results suggested that spatial working memory deficit may be a characteristic of autism, and may be distributed across the general population. Furthermore, we speculated a correlation between spatial working memory and visual perception in children with ASD and general population. The future study should be focused on the detailed mechanisms of the relationship between spatial working memory and visual perception.

## Supporting information

**S1 Data.**
(XLSX)

**S1 File. STROBE statement—checklist of items that should be included in reports of observational studies.**
(DOCX)

## Acknowledgments

We would like to thank all the families for generously donating their time so that this study could be possible. We are also grateful to the research staffs for their dedication to this project.

## Author Contributions

**Conceptualization:** Yi Huang.

**Data curation:** Xiao Hu, Pingyuan Yang, Kuifang Guo, Yi Huang.

**Formal analysis:** Yi Huang.

**Funding acquisition:** Mingjing Situ, Yi Huang.

**Investigation:** Manxue Zhang, Jian Jiao, Xiao Hu, Pingyuan Yang, Yan Huang, Mingjing Situ, Kuifang Guo, Jia Cai.

**Methodology:** Manxue Zhang, Jian Jiao, Xiao Hu, Kuifang Guo, Jia Cai, Yi Huang.

**Project administration:** Kuifang Guo, Yi Huang.

**Resources:** Manxue Zhang, Jian Jiao, Xiao Hu, Pingyuan Yang, Yan Huang, Mingjing Situ, Kuifang Guo, Yi Huang.

**Software:** Yan Huang.

**Supervision:** Mingjing Situ.

**Validation:** Jia Cai.

**Visualization:** Jia Cai.

**Writing – original draft:** Manxue Zhang.

**Writing – review & editing:** Manxue Zhang, Yi Huang.

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
