## [Decision Letter · Decision Letter 0]

27 Apr 2020

PONE-D-20-04238

Exploring the spatial working memory and visual perception in autism children and general population with high autism-like traits

PLOS ONE

Dear Pro. Huang,

Thank you for submitting your manuscript to PLOS ONE. After careful consideration, we feel that it has merit but does not fully meet PLOS ONE’s publication criteria as it currently stands. Therefore, we invite you to submit a revised version of the manuscript that addresses the points raised during the review process.

Minor methodological and expression details should be corrected and some references must be included, according to the reviewers' remarks. Particularly, authors should further elaborate the Discussion, incluiding a statement explaining how potential bias were addressed. In addition, an effort for the use of including language is required along the manuscript (e.g. children with ASD instead of ASD children, autistic children or ASD).

We would appreciate receiving your revised manuscript by Jun 07 2020 11:59PM. To enhance the reproducibility of your results, we recommend that if applicable you deposit your laboratory protocols in protocols.io, where a protocol can be assigned its own identifier (DOI) such that it can be cited independently in the future. For instructions see: http://journals.plos.org/plosone/s/submission-guidelines#loc-laboratory-protocols

We look forward to receiving your revised manuscript.

Kind regards,

Inmaculada Riquelme

Academic Editor

PLOS ONE

2. In your Methods section, please provide additional information about the participant recruitment method and the demographic details of your participants for both the case-control study and the community population study. Please ensure you have provided sufficient details to replicate the analyses such as:

a) the recruitment date range (month and year),

b) a description of any inclusion/exclusion criteria that were applied to participant recruitment,

c) a table of relevant demographic details,

d) a specific description of how participants were recruited online for the case-control study

e) descriptions of the specific locations where participants were recruited (names of the three primary schools for the community population study).

3. Please provide a sample size and power calculation in the Methods, or discuss the reasons for not performing one before study initiation.

4.  Please include additional information regarding the questionnaires or scales used in the study (ADOS-G, WISC-III, Beery VMI and SVM and ensure that you have provided sufficient details that others could replicate the analyses. For instance, if you developed a questionnaire as part of this study and it is not under a copyright more restrictive than CC-BY, please include a copy, in both the original language and English, as Supporting Information.

Reviewers' comments:

Reviewer's Responses to Questions

**Comments to the Author**

1. Is the manuscript technically sound, and do the data support the conclusions?

Reviewer #1: Yes

Reviewer #2: Yes

2. Has the statistical analysis been performed appropriately and rigorously? 

Reviewer #1: Yes

Reviewer #2: Yes

3. Have the authors made all data underlying the findings in their manuscript fully available?

Reviewer #1: Yes

Reviewer #2: Yes

4. Is the manuscript presented in an intelligible fashion and written in standard English?

Reviewer #1: Yes

Reviewer #2: Yes

5. Review Comments to the Author

Reviewer #1: 1. As the unbalanced gender ratios in ASD and TD group is a reckoned limitation, what has been done to address this?

2. In the discussion, 2nd paragraph, it was suggested that the lack of significant correlation between SWM and AQ could be explained by the heterogenous nature of ASD. Would the authors elaborate on this?

3. In the discussion, 2nd paragraph, it was postulated that the inconsistent SWM findings in subjects with high ALT was due to the lack of objective indicators of ALT. Do the authors mean that the ALTs they identify are not actually ALT? Such that the subsequent conclusions can not really be drawn?

4. On page 10, first paragraph, would the authors elaborate on the statement 'it enlightens that spatial working memory is a specifically cognitive elements in patients with ASD'. Following that, it was mentioned that SWM training might improve the performance of autism and alleviate the severity of the symptoms. Would the authors provide a reference for this?

5. The authors failed to provide an explanation on the inconsistent findings of normal visual processing skills in subjects with high ALT as compared to the existing literature.

6. The presence of correlation between SWM and visual perception in ASD and in the general population, and the lack of such correlation in the TD group was explained by ‘the interaction between cognitive functions’ as on page 11 line 11. Can the authors elaborate on this?

7. The conclusion suggested that the study results confirmed the deficits in neurocognitive functioning are autism-like traits and have a continuous distribution in the population. I wonder if the correlation analyses done in the population study can confer such a finding.

8. There are some typos (second sentence on page 4, second paragraph of discussion "which might be explainED") and the 1st sentence of the 3rd paragraph of P.10 was hard to read.

9. On page 9, second paragraph, the reference 46 was cited to support that idea that unaffected siblings showed normal SWM in high ALT. It was misleading as the study only compared HF-ASD, HF-ASD siblings and controls without assessment of autistic traits in the non-ASD groups.

Reviewer #2: Thank you for the opportunity to review this manuscript, in which the authors analyze spatial working memory and visual perception in children with Autism Spectrum Disorder and general population with high autism-like traits. This is a very interesting topic.

Despite the interest of the paper, I have some comments and recommendations for the authors.

First of all, the title refers to “autism children”, but using the expression “children with autism spectrum disorder” may be more appropriate.

Introduction

When they say “Previous researches had provided inconclusive evidence about whether children with ASD are impaired in working memory. Many investigations suggested working memory impairment in ASD”, the authors should be more precise about the age range they are referring to: do they include also preschoolers? Or only school age children? Do they include adolescents in their analysis?

When describing the results of the meta-analysis by Wang et al (2017), the authors should specify the age group on which this meta-analysis is focused, executive functions change deeply during development, the age group considered could have an important meaning to the inconsistent results in the field.

The authors discuss theories about the psychopathological mechanisms of autism, references are needed for these theories.

When they describe the aims of the study, the authors mention the clarification of inner mechanisms of ASD, but these are difficult to explore without a longitudinal analysis.

Materials and methods

Participants

Using DSM-5 diagnostic criteria that include descriptive specifiers for a more precise, and appropriate, nomenclature could be a better choice than the general distinction between low and high functioning individuals.

Age group of the 52 participants (minimum and maximum) should be specified. It is not clear what the iq value reported refers to, is it mean IQ? In this case, the authors should specify it and add sd and minimum and maximum IQ.

The authors should add information about the demographic features of the second sample, such as age and sex. There should be a detailed description of the sample in this section.

Measures

The AQ factors should be briefly described to allow a better understanding of the instrument.

Results

Part 1 – correlations among Neuropsychological Tests and Symptom Severity

Understanding mechanisms is only possible with longitudinal data, cross sectional data can provide associations, but not cause. It would be better if the authors did not imply that Pearson correlations can detect inner mechanisms of autism.

Part II – Degrees of Autism-like traits.

It is not clear here whether there was a matching procedure or if the authors mean that the groups did not significantly differ in age and iq. This should be clarified.

Discussion

When the authors mention the inconsistent results due to different focus points (page 10, line 6), references should be added.

Page 10, line 18: the expression broader autism phenotype (BAP) is usually referred to individuals somehow related to people diagnosed with ASD which is not the case here. BAP can be defined as a milder expression of the underlying genetic liability for autism, as present in relatives of individuals with ASD. I am not sure this is the most appropriate expression to use here.

Page 10, line 25: this sentences should be rephrased, as its meaning is not very clear.

Page 10, line 29: I think the authors meant typical development, instead of typically

Page 10, line 37: “Look it at the other way”: this sentences should be rephrased

Page 11, line 13: “the relationship that cannot be repeated might because of the samples and the measures”: a verb is missing after might

I think the study has a solid design and interesting conclusions and implications. Nevertheless, it could be strengthened by adding the clarifications and modifications required.

6. PLOS authors have the option to publish the peer review history of their article (what does this mean?). If published, this will include your full peer review and any attached files.

Reviewer #1: Yes: Lai Chun Lun Eric

Reviewer #2: No

---

## [Author Response · Author response to Decision Letter 0]

25 May 2020

Dear Editor,

Thank you for your valuable comments on our manuscript. We have revised the manuscript according to the reviewers’ comments and carefully proofread it to minimize errors.

Our point-to-point response to the reviewers’ comments are given below. 

Part Ⅰ：

Thank you for your valuable comments on our manuscript. We are sorry for all the mistakes. We have written the following information in the manuscript.

1.Please ensure that your manuscript meets PLOS ONE's style requirements, including those for file naming.  

Thanks. We have checked the style requirements.

2. In your Methods section, please provide additional information about the participant recruitment method and the demographic details of your participants for both the case-control study and the community population study. Please ensure you have provided sufficient details to replicate the analyses such as:

a) the recruitment date range (month and year),

page6 line5 during the period from September 2016 to June 2018.

b) a description of any inclusion/exclusion criteria that were applied to participant recruitment,

page 7 second paragraph line3 The exclusion criteria of our study included: a history of psychotic disorder, severe head injury, syndromic genetic disorder associated with autism (e.g. fragile X syndrome), intellectual disability (i.e. IQ <70), Tourette’s disorder, attention deficit hyperactivity disorder, any other medical condition significantly affecting brain function; unable to cooperate to complete the test.

c) a table of relevant demographic details,

 we have added Table 1(page10) and Table 3 (page13)to show demographic details of case-control study and population study respectively.

d) a specific description of how participants were recruited online for the case-control study

page6 line6 Our online recruitment advertises were on the social platform, WeChat official account and were distributed over a period of time to recruit more volunteers.

e) descriptions of the specific locations where participants were recruited (names of the three primary schools for the community population study).

Page 6 second paragraph line6 Pi Xian is a county with 750,000 population consists of 13 districts. The population study was conducted in Pi Xian county, Xi Pu district from October 2016 to January 2017. Two thousand nine hundred ninety-four school aged students in three public primary schools -Si Yuan experimental primary school, Xi Pu experimental primary school, and Xi Pu foreign language primary school- were included in our study which can stand for the characteristics of the children population of the district.

3. Please provide a sample size and power calculation in the Methods, or discuss the reasons for not performing one before study initiation.

Page 6 second paragraph line12 According to the sample size calculation formula N= (UαS/δ) 2, (Uα is the U value corresponding to the inspection level α, set as Uα=1.96; S is the standard deviation, δ is the allowable error; assume S / δ = 5), the estimated sample size is N= 99. Considering the loss of interview and quality control, we chose all the students in the population study. 

4.  Please include additional information regarding the questionnaires or scales used in the study (ADOS-G, WISC-III, Beery VMI and SVM and ensure that you have provided sufficient details that others could replicate the analyses. For instance, if you developed a questionnaire as part of this study and it is not under a copyright more restrictive than CC-BY, please include a copy, in both the original language and English, as Supporting Information.

Thank you for your valuable comments on our manuscript.

Autism Spectrum Quotient (AQ), additional information: 

Page 7 third paragraph line4 AQ-35 Chinese version consisted of five tightly semantically coherent subscale constructs, named as Socialness, Mindreading, Patterns, Attention to Details and Attention Switching. The items clustered in a way that depicted distinctive dimensions of ASD symptomatology.

Autism Diagnostic Interview-Revised (ADI-R) and Autism Diagnostic Observation Scale-General (ADOS-G), additional information:

Page 8 line5 The Chinese ADI-R was approved by the World Psychological Association in 2007 (56). The ratings were based on an assessment under current conditions and under the most severe state at 4–5 years, as recalled by the caregivers. It covers most developmental and behavioral aspects of ASD, including reciprocal social interaction, communication, and repetitive behaviors and stereotyped patterns. The ADOS-G is a semi-structured, standardized assessment of social interaction, communication, play, and imaginative use of materials (57). It has four different units for different levels of development and verbal abilities. ADOS-G provides scores that are distinct in three domains: language and communication score, reciprocal social interaction scores and total scores.

Chinese Version of the Wechsler Intelligence Scale (short-version), additional information:

Page 8 third paragraph line4 For the present investigation, FSIQ was measured using the WISC–III.

Spatial Working Memory (SWM), additional information:

Page 9 line1 Touching any box in which a blue token has already been found is an error. The number of boxes is gradually increased from three to eight boxes. The color and position of the boxes used are changed from trial to trial to discourage the use of stereotyped search strategies.

The Developmental Test of Visual-Motor Integration (Beery VMI), additional information:

Page 9 second paragraph line4 Considering the longitudinal nature of the parent project, the 3-5th editions of the Beery VMI were used during the various stages of data collection. The stimuli in the various editions did not change.

Part Ⅱ

Reviewer #1: 

Q1. As the unbalanced gender ratios in ASD and TD group is a reckoned limitation, what has been done to address this?

Thanks for your valuable comments. We are sorry about the mistake, and we tried to address the problem as follows. To minimize the effect of unmatched IQ, age, and gender ratio between groups, the ANCOVA was conducted with IQ, gender, and age as covariant. 

Table2. Performance differences on Spatial Working Memory (SWM) and Berry-VMI between ASD and control.

Q2. In the discussion, 2nd para

graph, it was suggested that the lack of significant correlation between SWM and AQ could be explained by the heterogenous nature of ASD. Would the authors elaborate on this?

Thanks for your valuable comments. We worked to discuss this issue as follows.

Page15 third paragraph Line8 On contrary, we found that the association between SWM and AQ was not statistically significant. It may be related to different age ranges. Previous studies were conducted in a wide range of age from children to adolescents, and our study was conducted in children aged from 6 to 12 years. Secondly, the tools used are different. A questionnaire, the Behavior Rating Inventory of Executive Function, BRIEF was used in the previous study; however, the present study used CANTAB which was a common and reliable measure. The BRIEF is not specific enough to capture separate EFs based on the subscales(66). Furthermore, symptoms may be affected by many factors, and the effect of cognitive function on symptoms may be related to other mediating factors. Interaction between cognitive functions may also have an impact on the results that SWM impairment found in ASD and high ALTs is not associated with overall level of autism symptom severity measured with the AQ. Another possible explanation for this pattern could be that spatial working memory is related to aspects of ASD that may not be covered well by the AQ. Additional studies using larger samples of participants in all age periods that combine laboratory and informant-based measures are necessary to clarify this issue.

Q3. In the discussion, 2nd paragraph, it was postulated that the inconsistent SWM findings in subjects with high ALT was due to the lack of objective indicators of ALT. Do the authors mean that the ALTs they identify are not actually ALT? Such that the subsequent conclusions can not really be drawn?

Thanks for pointing this out. We are sorry about that we mentioned the reasons of the inconsistent in an inappropriate way. We have elaborated on this as follows.

Page16 Line9 However, recently one study conducted in unaffected siblings, showed spatial working memory wasn’t a broader autism phenotype of ASD [48]. The previous study didn’t identify spatial working memory as broader autism phenotype of ASD, which may be explained by the small sample size and the limitation of the term ‘broader autism phenotype’; and in our study we applied AQ to measure ALTs.

Q4. On page 10, first paragraph, would the authors elaborate on the statement 'it enlightens that spatial working memory is a specifically cognitive elements in patients with ASD'. Following that, it was mentioned that SWM training might improve the performance of autism and alleviate the severity of the symptoms. Would the authors provide a reference for this?

Thanks for pointing this out. We are sorry about the inappropriate expressions.

Page16 Line18 Investigating executive function components that contribute to the severity of autistic symptomology provides insight for the development of effective interventions for individuals with ASD. Previous study proposed that the working memory trains improved classroom performances of children with ASD (68). We speculated that spatial working memory was an autism-like trait; and professional spatial working memory training might alleviate the severity of the symptoms.

Q5. The authors failed to provide an explanation on the inconsistent findings of normal visual processing skills in subjects with high ALT as compared to the existing literature.

Thanks for your valuable comments. we have rewritten the discussion of this issue.

Page 16 second paragraph line3 In our study, the visual perception function deficit was found in ASD group when compared with typically developing children. Previous studies have found that individuals with autism have visual perception dysfunctions, which was predictor to the severity of symptoms (70-75). Except that Green et al. did not repeat the results in children with ASD (64). However, the population study results indicated that higher trait group behave comparable of lower trait group. This is at odds with previous research suggesting that people with high autistic traits have impaired visual illusions (76), and visual search(77). It should be noted that the AQ also correlates with IQ(78), and it could be IQ that produces the no significant results of high trait group. In addition, the visual perception inconsistency of the ALTs prompted us to carry out better visual psychophysics using more sensitive measures of perception in general population with various degrees of autistic traits. Future brain mapping studies could provide additional insight into the neural underpinnings of how visual perception might and might not be affected in autism.

Q6. The presence of correlation between SWM and visual perception in ASD and in the general population, and the lack of such correlation in the TD group was explained by ‘the interaction between cognitive functions’ as on page 11 line 11. Can the authors elaborate on this?

Thanks for your valuable comments. we have rewritten the discussion of this issue.

Page 18 line3 The present study found that visual perception deteriorated as working memory performance decreased in the ASD group. While the relationship didn’t repeat in control group, possibly because copying geometric figures in the VMI tasks was not so hard for typically developing children to remember; at the same time children could check the geometric figures again if they forgot them. Also, the small sample size may have an impact on the results of control group, and we tried to address it by community population study with a larger sample size, and results of the general population children suggested a positive correlation between SWM and visual perception. Researches in preschool children suggested working memory has no impact on the VMI skills[40], the age range might explain the controversy. Further study was needed to explore mechanism of the relationship between working memory and visual perception using more meticulous measures. 

Q7. The conclusion suggested that the study results confirmed the deficits in neurocognitive functioning are autism-like traits and have a continuous distribution in the population. I wonder if the correlation analyses done in the population study can confer such a finding.

Thanks for pointing this out. We are sorry about the inappropriate expressions. We have rewritten the sentence.

Page 19 second paragraph line2 Our results suggested that spatial working memory deficit was a characteristic of autism, and may be distributed across the general population. Furthermore, we speculated a correlation between spatial working memory and visual perception in children with ASD and general population.

Q8. There are some typos (second sentence on page 4, second paragraph of discussion "which might be explainED") and the 1st sentence of the 3rd paragraph of P.10 was hard to read.

Thanks for pointing this out. We are sorry about the mistakes we have made.

1.second sentence on page 4: Visual perception and executive function were studied in the same task, The results suggest that visual perception is dissociable with executive function(39).

Page 5 line4 Karisa and his team, using a sophisticated measure that tested visual processing and executive function, found that visual perception is dissociable with executive function in individuals with various degree of ALTs(41).

2.second paragraph of discussion "which might be explained

we have rewritten the paragraph, and the sentence has been deleted already.

3.1st sentence of the 3rd paragraph of P.10: Traditional case-control studies are difficult to find the characteristics of quantitative phenotypes such as ASD, while population study can overcome this shortcoming

Page 17 second paragraph line1 It is challenging for traditional case-control studies to invest diseases that distributed normally in population. However, population study is just helpful to explore the diseases with a continuous distribution of characteristics in the population (79).

Q9. On page 9, second paragraph, the reference 46 was cited to support that idea that unaffected siblings showed normal SWM in high ALT. It was misleading as the study only compared HF-ASD, HF-ASD siblings and controls without assessment of autistic traits in the non-ASD groups.

Thank you for your valuable comments on our manuscript. We are sorry about the mistake. What we mentioned may be ‘broader autism phenotype’ but not high ALT. 

Page 16 line9 However, recently one study conducted in unaffected siblings, showed spatial working memory wasn’t a broader autism phenotype of ASD [48]. The previous study didn’t identify spatial working memory as broader autism phenotype of ASD, which may be explained by the small sample size and the limitation of the term ‘broader autism phenotype’; and in our study we applied AQ to measure ALTs.

Part Ⅲ

Reviewer #2: 

Thank you for the opportunity to review this manuscript, in which the authors analyze spatial working memory and visual perception in children with Autism Spectrum Disorder and general population with high autism-like traits. This is a very interesting topic.

Despite the interest of the paper, I have some comments and recommendations for the authors.

Q1. First of all, the title refers to “autism children”, but using the expression “children with autism spectrum disorder” may be more appropriate.

Thank you for your valuable comments on our manuscript. We have changed the title as ‘Exploring the spatial working memory and visual perception in autism children with Autism Spectrum Disorder and general population with high autism-like traits’, and in the manuscript all the words refer to autism，autistic children, are using ‘children with ASD’ instead.

Introduction

Q2. When they say “Previous researches had provided inconclusive evidence about whether children with ASD are impaired in working memory. Many investigations suggested working memory impairment in ASD”, the authors should be more precise about the age range they are referring to: do they include also preschoolers? Or only school age children? Do they include adolescents in their analysis?

When describing the results of the meta-analysis by Wang et al (2017), the authors should specify the age group on which this meta-analysis is focused, executive functions change deeply during development, the age group considered could have an important meaning to the inconsistent results in the field.

Thank you for your valuable comments on our manuscript. We have carefully reread the references and described the age ranges of these researches in the manuscript. 

Page4 line1 Investigations of school aged children and adolescents suggested working memory impairment in ASD (21, 22). Working memory deficit is associated with symptoms of children and adolescents with ASD such as communication and socialization impairments (23, 24), restrictive and repetitive interests and behaviors (25). While some researches indicated normal working memory in preschoolers and adults with ASD(26-28). The mixed results indicated that adolescents with ASD showed remarkable working memory dysfunction, and indicated that age had important influences in the field. A Meta-Analysis (29) of working memory in individuals ranged from school-age children to adults suggested that spatial working memory was more severely impaired than verbal working memory.

Q3. The authors discuss theories about the psychopathological mechanisms of autism, references are needed for these theories.

Thanks for pointing this out. We are sorry about this mistake; we have attached the references:32-33

Page4 second paragraph line2 There have been many theories about the psychopathological mechanisms of ASD, such as ‘executive function deficit’, and ‘weak central coherence’[32, 33]

Q4. When they describe the aims of the study, the authors mention the clarification of inner mechanisms of ASD, but these are difficult to explore without a longitudinal analysis.

Thanks for pointing this out. We agree with you that the correlation analysis couldn’t stand for the relationship of cause and result, so it is unsuitable to mention inner mechanisms.

Page5 third paragraph line5 Furthermore, we detected the correlations between visual perception and spatial working memory. 

Materials and methods

Participants

Q5. Using DSM-5 diagnostic criteria that include descriptive specifiers for a more precise, and appropriate, nomenclature could be a better choice than the general distinction between low and high functioning individuals.

Thanks for pointing this out. We have changed ‘high functioning ASD’ to ‘ASD without intellectual disability comorbidity’

Page5 fourth paragraph line2 We recruited ASD without intellectual disability comorbidity

Q6. Age group of the 52 participants (minimum and maximum) should be specified. It is not clear what the iq value reported refers to, is it mean IQ? In this case, the authors should specify it and add sd and minimum and maximum IQ.

Thank you for your valuable comments on our manuscript. We are sorry about this mistake. We have added those demographic features in the manuscript. 

Page5 fourth paragraph line2 We recruited ASD without intellectual disability comorbidity(n=52, mean age=9.23 years, SD=3.35 years, Min–Max=4-17 years; mean IQ=95.27,SD=19.18 , Min–Max=70-132; Male/Female=37/7) and typically developed control children (n=32, mean age=10.63 years, SD=3.15，Min–Max=6-16years; IQ=108.22, SD=13.94 , Min–Max=81-136; Male/Female=16/16). (see in table1.)

Q7. The authors should add information about the demographic features of the second sample, such as age and sex. There should be a detailed description of the sample in this section.

Thanks for pointing this out. We have added those demographic features in the manuscript. 

Page7 second paragraph line7：Finally, the demographic features of the population study participants were as follows, higher trait group(n=115, mean age=8.35 years, SD=1.42 years, Min–Max=6-11 years; mean IQ=117.37, SD=12.25, Min–Max=86-139; Male/Female=44/71), lower trait group(n=105, mean age=8.03 years, SD=1.50 years, Min–Max=6-11 years; mean IQ=116.16,SD=10.74 , Min–Max=82-137; Male/Female=42/63).(see in table 3.)

Measures

Q8. The AQ factors should be briefly described to allow a better understanding of the instrument.

Thank you for your valuable comments on our manuscript. We agree with you that more detailed about the AQ subscales should be mentioned. 

Page7 third paragraph line4：AQ-35 Chinese version consisted of five tightly semantically coherent subscale constructs, named as Socialness, Mindreading, Patterns, Attention to Details and Attention Switching. The items clustered in a way that depicted distinctive dimensions of ASD symptomatology.

Results

Q9. Part 1 – correlations among Neuropsychological Tests and Symptom Severity

Understanding mechanisms is only possible with longitudinal data, cross sectional data can provide associations, but not cause. It would be better if the authors did not imply that Pearson correlations can detect inner mechanisms of autism.

Thanks for pointing this out. We are sorry for the mistake we have made. We agree that it is not appropriate to say that correlation analysis can imply inner mechanism, and such expression has been deleted

Page12 fourth paragraph Line3 The results showed the correlation between SWM and visual perception in children with ASD did not repeat in typically developing controls. 

Part II – Degrees of Autism-like traits.

Q10. It is not clear here whether there was a matching procedure or if the authors mean that the groups did not significantly differ in age and iq. This should be clarified.

Thank you for your valuable comments on our manuscript. In the present study we mean that ‘Higher trait group and lower trait group did not significantly differ in age and IQ.’ 

Discussion

Q11. When the authors mention the inconsistent results due to different focus points (page 10, line 6), references should be added.

Thanks for pointing this out. We are sorry for the mistake we have made. we have attached the references,13, 15-17, 70

Page16 second paragraph line1 However, previous studies on visual processing had inconsistent results due to different focus points (13, 15-17, 70).

Q12. Page 10, line 18: the expression broader autism phenotype (BAP) is usually referred to individuals somehow related to people diagnosed with ASD which is not the case here. BAP can be defined as a milder expression of the underlying genetic liability for autism, as present in relatives of individuals with ASD. I am not sure this is the most appropriate expression to use here.

Thanks for pointing this out. We have rewritten the paragraph and the sentence has been deleted already.

Q13. Page 10, line 25: this sentence should be rephrased, as its meaning is not very clear.

Thanks for pointing this out. We are sorry for the mistake. we have rephrased the sentences as follows. 

Page 17, second paragraph line1 It is challenging for traditional case-control studies to invest diseases that distributed normally in population. However, population study is just helpful to explore the diseases with a continuous distribution of characteristics in the population(79).

Page 10, line 29: I think the authors meant typical development, instead of typically

Thanks for pointing this out. We are sorry for the mistake, and we have corrected the error.

 Page 10, line 37: “Look it at the other way”: this sentence should be rephrased

Thanks for pointing this out. We are sorry for the mistake, and we have rephrased the sentence.

Page 17, second paragraph line17 Besides, the present study offered support to the approaches that regard autistic-like traits as a continuous variable in general population rather than qualitative variable.

Page 11, line 13: “the relationship that cannot be repeated might because of the samples and the measures”: a verb is missing after might

Thanks for pointing this out. We are sorry for the mistake. We have rephrased this sentence as follows.

Page 18, line11 Researches in preschool children suggested working memory has no impact on the VMI skills[40], the age range might explain the controversy.

I think the study has a solid design and interesting conclusions and implications. Nevertheless, it could be strengthened by adding the clarifications and modifications required.

Thanks for your valuable comments!

---

## [Decision Letter · Decision Letter 1]

18 Jun 2020

Exploring the spatial working memory and visual perception in children with Autism Spectrum Disorder and general population with high autism-like traits

PONE-D-20-04238R1

Dear Dr. Huang,

We’re pleased to inform you that your manuscript has been judged scientifically suitable for publication and will be formally accepted for publication once it meets all outstanding technical requirements.

Kind regards,

Inmaculada Riquelme

Academic Editor

PLOS ONE

Additional Editor Comments (optional):

Reviewers' comments:

Reviewer's Responses to Questions

**Comments to the Author**

1. If the authors have adequately addressed your comments raised in a previous round of review and you feel that this manuscript is now acceptable for publication, you may indicate that here to bypass the “Comments to the Author” section, enter your conflict of interest statement in the “Confidential to Editor” section, and submit your "Accept" recommendation.

Reviewer #1: All comments have been addressed

2. Is the manuscript technically sound, and do the data support the conclusions?

Reviewer #1: Yes

3. Has the statistical analysis been performed appropriately and rigorously? 

Reviewer #1: Yes

4. Have the authors made all data underlying the findings in their manuscript fully available?

Reviewer #1: Yes

5. Is the manuscript presented in an intelligible fashion and written in standard English?

Reviewer #1: Yes

6. Review Comments to the Author

Reviewer #1: (No Response)

7. PLOS authors have the option to publish the peer review history of their article (what does this mean?). If published, this will include your full peer review and any attached files.

Reviewer #1: Yes: LAI CHUN LUN ERIC

---

## [Editor Report · Acceptance letter]

26 Jun 2020

PONE-D-20-04238R1 

Exploring the spatial working memory and visual perception in children with Autism Spectrum Disorder and general population with high autism-like traits 

Dear Dr. Huang:

I'm pleased to inform you that your manuscript has been deemed suitable for publication in PLOS ONE. Congratulations! Your manuscript is now with our production department. 

Kind regards, 

on behalf of

Dr. Inmaculada Riquelme 

Academic Editor

PLOS ONE